# Slight Shading Stress at Seedling Stage Does not Reduce Lignin Biosynthesis or Affect Lodging Resistance of Soybean Stems

**Bingxiao Wen** , **Yi Zhang, Sajad Hussain** , **Shan Wang, Xiaowen Zhang, Jiayue Yang, Mei Xu, Sisi Qin, Wenyu Yang \* and Weiguo Liu \***

Institute of Ecological Agriculture, Sichuan Agricultural University/Sichuan Engineering Research Center for Crop Strip Intercropping System, Chengdu 611130, China; wenbingxiao@outlook.com (B.W.); 2019201010@stu.sicau.edu.cn (Y.Z.); Hussainsajjad456@Yahoo.com (S.H.); WangShan9861@163.com (S.W.); zhangxiaowen@sicau.edu.cn (X.Z.); zxc602087577@163.com (J.Y.); XuMei2094@163.com (M.X.); QinSisi327@163.com (S.Q.)
\* Correspondence: mssiyangwy@sicau.edu.cn (W.Y.); lwgsy@126.com (W.L.)

**Abstract:** Shade is widespread in agricultural production and affects lignin biosynthesis and lodging resistance of crops. We explored the effects of shade intensity on lignin biosynthesis and lodging resistance at the physiological and molecular levels in two soybean cultivars (Nandou12 and E93) with different shade tolerance under four progressively severe shade treatments, S0–S3 (S0: no shade, S1: slight shade, S2: moderate shade, S3: heavy shade). Our results showed no significant difference in breaking strength of the two cultivars under S1 and S0 treatments, with no prominent decrease in the lodging resistance index. The activity of lignin biosynthesis rate-limiting enzymes phenylalanine ammonia-lyase (PAL), peroxidase and cinnamyl alcohol dehydrogenase (CAD), which were considerably related to the two lodging resistance indexes above, was not significantly decreased by slight shade, while 4-coumaric acid ligase (4CL) activity was increased. Most genes involved in lignin biosynthesis were not significantly down-regulated by slight shade (S1) compared to S0, while p-coumarate 3-hydroxylase (*C3H*), 4-coumaric acid ligase (*4CL*) and laccase (*LAC*) genes were upregulated. Under heavy shade (S3), enzyme activity and gene expression associated with lignin synthesis in both soybean cultivars were strongly inhibited; moreover, stem mechanical strength and lodging resistance were remarkably decreased compared with those under S0. These physiological and molecular changes suggested that applicable shade levels do not significantly affect the mechanical strength and lodging resistance of soybean stem. Exploiting the lodging resistance potential of existing soybean cultivars was an effective and efficient way to address yield reduction caused by lodging in intercropped soybeans.

**Keywords:** lodging; lodging resistance; lignin; enzyme activity; gene expression

## 1. Introduction

Light is an indispensable but relatively difficult to control factor in agricultural production, because it mainly comes from solar radiation [1]. For this reason, meteorological factors determine the average annual amount of radiation in a specific region. Furthermore, some areas lack light resources due to geographical factors. Shade environments are also widely found in various cultivation systems: there is interspecies shading in the dense planting model [2] and there is strong intraspecies shading in the intercropping system [3]. Moreover, in some interforest cropping systems, there are also shade environments where the shorter plants cannot escape [4].



Changes in quality and quantity of light trigger a shade avoidance response influencing many aspects of plant development, resulting in slender, weaker stems, reduced efficiency of the photosynthetic apparatus and susceptibility to lodging [5]. The physical fitness of the stem of a crop is crucial to its survival under shading stress. Soybean is widely cultivated worldwide. In southwestern and western China, soybean is often intercropped with maize for more efficient use of land resources and higher economic benefits. Similar to other non-shade plants, although general growth and development of soybean are affected by shade, the most significant manifestation is increased susceptibility to lodging. Lodging inhibits some physiological processes in soybean and greatly increases the risk of grain mildew [6], which adversely affects yield and quality.

Lignin is the second most abundant, hard, rigid and complex aromatic polymer in vascular plants and is an important constituent of cell walls [7]. It accounts for approximately 30% of organic carbon in plant biomass [8]. It plays an important role in enhancing rigidity to protect plants against pathogen attack and mechanical stress and enhances the strength of cell walls by crosslinking with cellulose and hemicellulose [9]. Stem mechanical strength is an essential agronomic feature which affects the normal development of crops, because lodging early in development leads to extensive losses [10]. Lignin is composed of three p-hydroxycinammyl alcohol precursors or monolignols, namely coniferyl, p-coumaryl and sinapyl alcohols. Their polymerization results in the hydroxyphenyl (H), guaiacyl (G) or syringyl (S) lignin units, respectively [11]. The enzymatic activities of phenylalanine ammonia-lyase (PAL), cinnamyl alcohol dehydrogenase (CAD), 4-coumaric acid ligase (4CL) and peroxidase [12] play a major role in lignin biosynthesis [13]. PAL catalyzes the conversion of L-phenylalanine ammonia from trans cinnamic acid and is the first rate-limiting enzyme in the lignin biosynthesis pathway [14]. The 4CL is the last enzyme in the phenylpropionic acid pathway, catalyzing conversion of 4-coumaric acid and other derivatives containing hydroxyl or methoxy styrenic acid to their corresponding esters with coenzyme A, a rate-limiting enzyme connecting the phenylpropionic acid pathway and the lignin biosynthesis pathway [15]. Evaluation of shade-tolerant and shade-susceptible genotypes confirms higher lignin contents in shade-resistant genotypes [16]. Enzymatic activities (PAL, CAD, 4CL and POD: peroxidase) involved in lignin biosynthesis are higher in shade-tolerant genotypes as compared to shade-susceptible genotypes [5,17]. Furthermore, rice genotypes with different levels of shade tolerance show consistent results, with shade being a detrimental and limiting factor in the lignin biosynthesis process. Through reducing the enzymatic activities of PAL, CAD, 4CL and POD, shade reduces lignin content resulting in weak stem physical strength [18].

Due to the inevitability of shading in intercropping systems, many studies have sought to mitigate adverse effects by breeding shade-tolerant soybean cultivars [19]. However, exploiting the shade-tolerance potential of existing cultivars and elucidating the tolerance of different genotypes to shade is a more economical and efficient way to solve this problem. In addition, some studies have suggested that the degree of adverse effects of shade on plants also depends on the intensity of shade and the stage of plant growth [20]. Therefore, we used four shade treatments (S0–S3) with different intensities to evaluate two cultivars with different performance under shade. We explored the effects of shade intensity on lignin biosynthesis and lodging resistance in soybean at the physiological and molecular levels. We found that severe shading significantly affected lodging resistance and lignin synthesis in soybean, while slight shading did not.

## 2. Materials and Methods

### 2.1. Experimental Site and Planting Material

The experiment was carried out at the experimental farm (30°97′ N, 103°81′ E; 647 m elevation) of Sichuan Agricultural University in Wenjiang, Chengdu, Sichuan Province of China. The experiment took place in the summer of 2017, from July to October. The two soybean cultivars selected for the experiment were 'Nandou12' (the main cultivar in southwest China with strong shade tolerance,

selected by Nanchong Academy of Agricultural Sciences, Sichuan province, China) and 'E93' (control cultivar: a shade intolerant cultivar).

## 2.2. Treatments and Experimental Details

The experiment followed a completely randomized block design with three replications. Black shade nets of appropriate density were used to set up four different shade treatments, namely S0 (no shade), S1 (slight, or 43% reduction in light intensity), S2 (moderate, or 58% reduction) and S3 (heavy, or 73% reduction). The sowing date was 3 July 2017; daily temperature and humidity are shown in Figure 1. Soybean seeds were planted in plastic pots, each 30 cm deep and 25 cm in diameter, filled with 10 kg of field soil: loamy, pH 6.7, total nitrogen content 2.1g kg$^{-1}$, magnesium content 21.3 g kg$^{-1}$, available phosphorus content 24.5 mg kg$^{-1}$, available potassium content 121 mg kg$^{-1}$, available nitrogen content 130 mg kg$^{-1}$. There was one seed per pot and 30 pots for each treatment; pots were placed in the field at the experimental farm. After germination, seeds were treated with different shading levels, and the shading net was removed before the initial flowering stage (R1). Every soybean plant was maintained until mature.

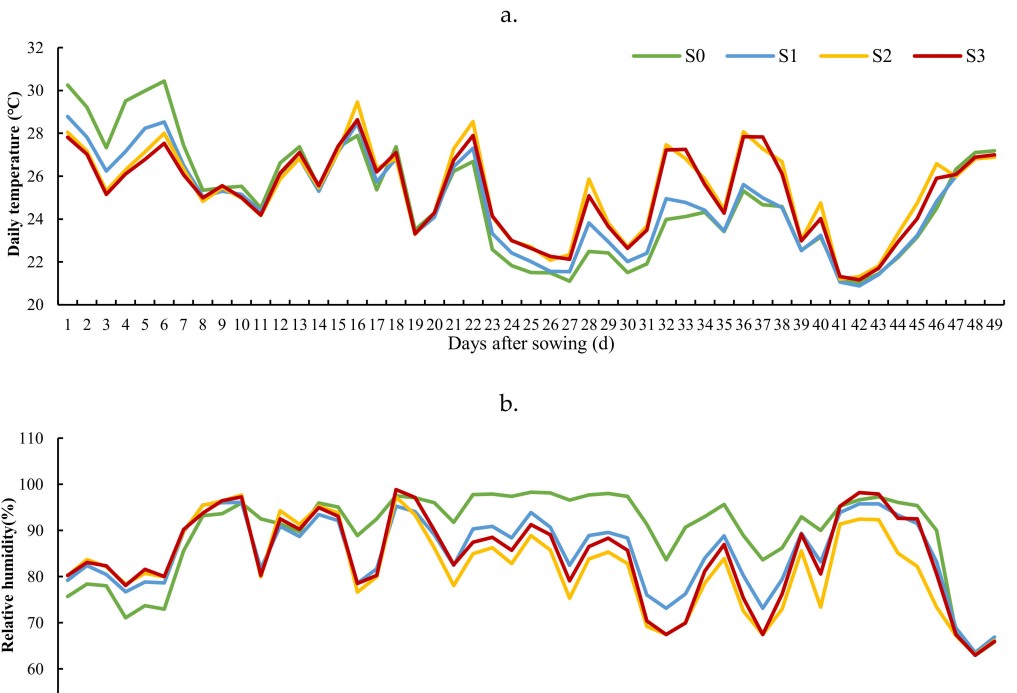

**Figure 1.** Daily temperature (**a**) and humidity (**b**) after sowing. The humidity and daily temperature under S0–S3 were measured by Meteo and the Bowen ratio system (Jiri Kucera, Brno, Czech Republic). S0: no shade, S1: slight, or 43% reduction in light intensity, S2: moderate, or 58% reduction, S3: heavy, or 73% reduction.

## 2.3. Sampling and Measurement

### 2.3.1. Indicators of Lodging Resistance

The whole aboveground part of each soybean plant was sampled at the V5 stage (five compound leaves fully unfolded). Plant height was measured with a straight ruler and stem diameter was measured with a Vernier caliper. Stem breaking strength was measured using a stem strength meter (YYD1, Zhejiang Pu Instrument Co. LTD, Hangzhou, China) and the index of lodging resistance was

calculated as follows: index of lodging resistance = (stem breaking strength × stem diameter)/(dry weight of aboveground part × main stem length).

### 2.3.2. Stem Anatomical Structure

Stem tissues were fixed with formalin-acetic acid-ethanol fixation solution (FAA), then dehydrated with ethanol, made transparent with xylene, soaked in melted paraffin wax for 30 min and placed in a container to cool naturally for fixation and embedding. The paraffin block was sectioned with a microtome (RM2235, Leica, Heidelberg, Germany), and a continuous wax strip was obtained for mounting and drying, which was flattened as much as possible. After drying, samples were dewaxed with xylene and ethanol of different concentrations (100%-0). Finally, stem sections were stained with Sarranine-Fast Green (GP1059, Servicebio, Wuhan, China), and the thickness of xylem and pith were observed using a digital microscope (Nikon 50i, Tokyo, Japan). The length of each tissue and cell was measured using Photoshop CC (Adobe Systems Software, San Jose, CA, USA).

### 2.3.3. Carbohydrates

Dried stem samples were ground to a powder and sifted through 60 mesh. Soluble sugar and sucrose contents were determined in 0.1g samples using the anthrone-sulfuric acid method [21], and starch content was measured using the resorcinol method [22]. Each measurement was repeated three times.

### 2.3.4. Lignin and Cellulose Content

A 5mg sample of soybean stem powder from each treatment was used to determine lignin and cellulose contents with commercially available kits (Qiyi Biological Technology, Shanghai, China). Each sample was tested three times.

### 2.3.5. Lignin Biosynthesis-Related Enzyme Activity

Powdered fresh samples (exactly 2 g) were extracted with pH 7.0 phosphate buffer. The activity of PAL, 4CL, POD and CAD was determined using enzyme activity assay kits (Shanghai Fu Life Industry Co., Ltd., Shanghai, China). Each sample was measured three times.

### 2.3.6. Gene Expression

Reverse-transcription polymerase chain reaction (PCR) assays were used to detect the expression of genes related to lignin metabolism in stems [23], including laccase (*LAC*), p-coumarate 3-hydroxylase (*C3H*), 4-coumarate: CoA ligase (*4CL*), phenylalanine ammonia-lyase (*PAL*), cinnamate 4-hydroxylase (*C4H*), cinnamyl alcohol dehydrogenase (*CAD*), caffeic acid O-methyltransferase (*COMT*), caffeoyl-CoA O-methyltransferase (*CCoAOMT*) and cinnamoyl-CoA reductase (*CCR*); primer sequences are given in Table S1. Total RNA was extracted from each treated stem tissue using a total RNA extraction kit (TIANGEN Biotech, Beijing, China). RNA concentrations were determined using a ND-2000 ultraviolet Spectrophotometer (Thermo Scientific, Waltham, MA, USA). ReverTra Ace qPCR RT Master (TOYOBO CO. Osaka, Japan) was used to transcribe the RNA extracted from each treatment into cDNA. Quantitative PCR using *GmACTIN1* as an internal reference gene was performed on a Quant Studio 6 Flex real-time PCR System (Thermo Fisher Scientific, Waltham, MA, USA); real-time detection was performed using Vazyme$^{TM}$ AceQ qPCR SYBR Green Master mix (vazyme biotech co., Nanjing, China), and data were analyzed and calculated using the 2-△△Ct method [24].

### 2.3.7. Statistical Analysis

Statistical analysis of the four environmental treatments, two cultivars and three replications for each parameter involved analysis of variance (ANOVA), which was carried out using SPSS ver. 19.0 (SPSS, Chicago, IL, USA). Probability values of less than 5% were considered significant. Tables and graphs were prepared using Microsoft Excel 2013 (Microsoft, Redmond, WA, USA).

## 3. Results

### 3.1. Effects of Shading Intensity on Phenotypic Indexes of Soybean Stems

As shown in Figure 2a, plant height of the two cultivars showed a similar increasing trend with increasing shade level. The height of 'E93' was greater than that of 'Nandou12' except under S3 treatment (heavy shade). Figure 2b shows the stem diameter of two soybean cultivars under different shade intensities. Under S1 treatment, the stem diameter of 'Nandou12' and 'E93' was 2.2% and 2.6% smaller, respectively, compared with that under S0 (control), which was not significant. However, under S2 and S3 treatments, the stem diameter of both cultivars was considerably lower than that under S0, with diameter under S2 greater than that under S3.

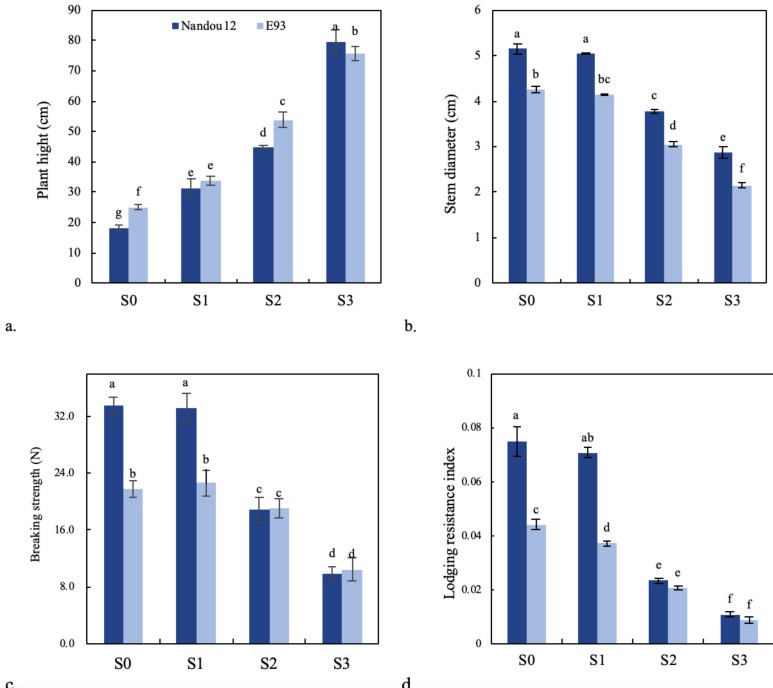

**Figure 2.** Effects of different shade degrees on plant height (**a**) and stem diameter (**b**) breaking strength (**c**) and lodging resistance index (**d**) of two cultivars of soybean. Different letters indicate statistically significant differences between treatments at $p < 0.05$ by Duncan's multiple range test. Vertical bars represent ± standard deviation (*n* = 3). S0: no shade, S1: slight, or 43% reduction in light intensity, S2: moderate, or 58% reduction, S3: heavy, or 73% reduction.

### 3.2. Effects of Shading Intensity on Breaking Strength and Lodging Resistance Index

Under our experimental conditions, compared with S0, the stem-breaking strength of the two soybean cultivars was not significantly lower under S1 treatment; it is worth noting that the stem-breaking strength of 'E93' increased slightly compared with that under S0, by 3.8% (Figure 2c). However, heavier shade caused a remarkable decline in stem-breaking strength. Under S0 and S1 treatments, the stem breaking strength of 'Nandou12' was much higher than that of 'E93', while the difference between the two cultivars under S2 and S3 treatments was not significant. The lodging resistance index of the two cultivars under S1 treatment was 5.8% and 15.6% lower, respectively, than that under S0 (Figure 2d). However, the decline was less than that seen in plants under S2 and S3 treatments. Moreover, the lodging resistance index of 'Nandou12' was much higher than that of 'E93' under these two shade levels, consistent with stem breaking strength.

### 3.3. Effects of Shading Intensity on the Proportion of Xylem in Soybean Stems

Table 1 shows that with increasing shade intensity, the xylem area and pith area of the two soybean cultivars showed a significant decreasing trend (S0 > S1 > S2 > S3), with xylem and pith area of 'Nandou12' greater than those of 'E93' under each treatment. There was no marked difference in the pith proportion of 'E93' under different shade levels (Figure 3a). Moreover, we observed no significant difference in the pith proportion of 'Nandou12' under S0 and S1; data for these two groups were significantly lower than those for S2 and S3. Of note, the xylem proportion of the two cultivars under S1 was significantly higher than that under S0 (by 3.44% and 1.04%, respectively), with the two cultivars showing a similar trend (Figure 3b).

**Table 1.** The area of xylem and pith of soybean stem under different shade degrees.

|  |  | Xylem Area (mm$^2$) | Pith Area (mm$^2$) |
|---|---|---|---|
| Nandou12 | S0 | 13.493 ± 0.013 a | 5.587 ± 0.008 c |
|  | S1 | 12.293 ± 0.009 b | 4.681 ± 0.005 b |
|  | S2 | 6.389 ± 0.013 e | 4.035 ± 0.007 e |
|  | S3 | 3.614 ± 0.005 g | 2.584 ± 0.004 g |
| E93 | S0 | 10.576 ± 0.020 c | 6.252 ± 0.12 a |
|  | S1 | 8.457 ± 0.005 d | 5.115 ± 0.008 d |
|  | S2 | 4.363 ± 0.019 f | 2.769 ± 0.002 f |
|  | S3 | 1.816 ± 0.011 h | 1.599 ± 0.003 h |

Different letters indicate statistically significant differences between treatments at $p < 0.05$ by Duncan's multiple range test. Values are the mean ± standard deviation (SD, $n = 3$). S0: no shade, S1: slight, or 43% reduction in light intensity, S2: moderate, or 58% reduction, S3: heavy, or 73% reduction.

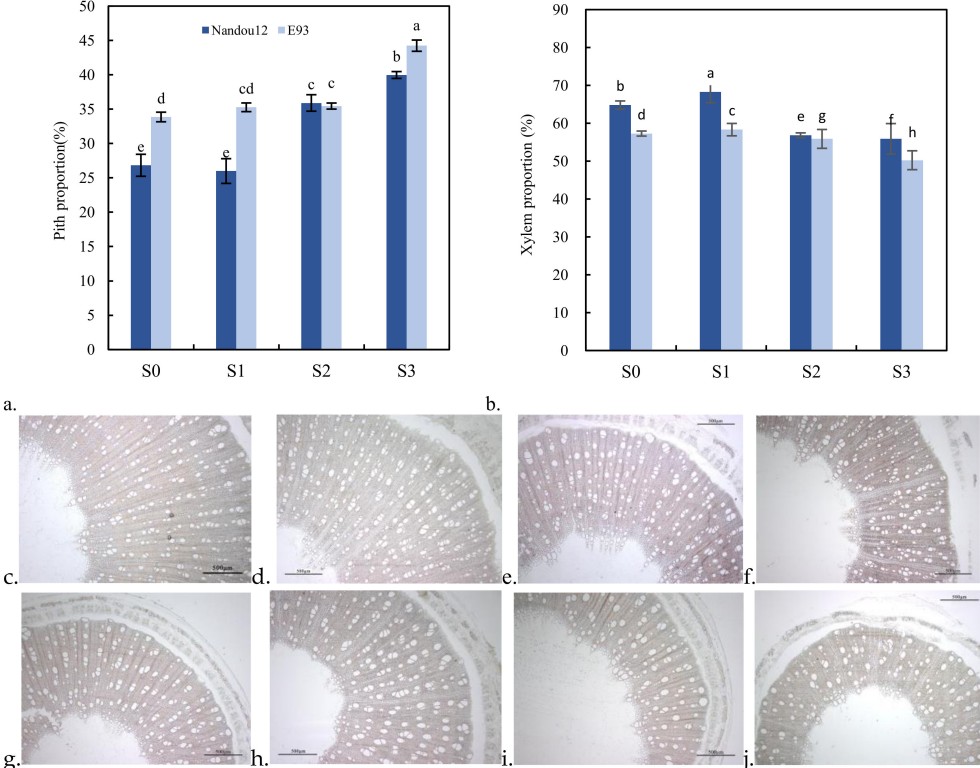

**Figure 3.** Effect of different shading levels on the proportion of soybean xylem. The proportion of xylem (**a**) and pith (**b**) of soybean stem under different shade degrees. Different letters indicate statistically significant differences between treatments at $p < 0.05$ by Duncan's multiple range test. Values are mean ± SD ($n = 3$). (**c**–**f**): anatomic structure of stem of Nandou12 under S0–S3, (**g**–**j**): anatomic structure of stem of E93 under S0–S3.

### 3.4. Effects of Shading Intensity on Non-Structural Carbohydrates in Stem

Non-structural carbohydrates in the stem of two cultivars under S1 and S2 were significantly higher than those under S0 due to the high shading level (Figure 4). The higher the shade intensity, the higher the accumulation of soluble sugar: S3 > S2 > S1 (Figure 4a); moreover, the soluble sugar content of 'Nandou12' under each treatment was higher than that of 'E93'. Sucrose content showed a slightly different trend; only treatment S3 caused a significant increase compared with S0. The sucrose accumulation in the two cultivars under treatment S3 was 36.10% and 38.16% higher than that of the control group, respectively (Figure 4b). Additionally, under S1 and S2, sucrose content in the stem of 'Nandou12' was higher than that in 'E93', but there was no significant difference between treatments S0 and S1. Starch accumulation also showed a trend of S3 > S2 >S1, and the lowest value was obtained under S1 (Figure 4c). 'Nandou12' contained less starch under three shaded treatments than under S0, while 'E93' contained more starch under S2 and S3. It is worth noting that there was no significant difference in soluble sugar or sucrose content between the S1 and S0 treatments, while starch content under S1 was considerably lower than that under S0; there was no significant difference between the two soybean cultivars.

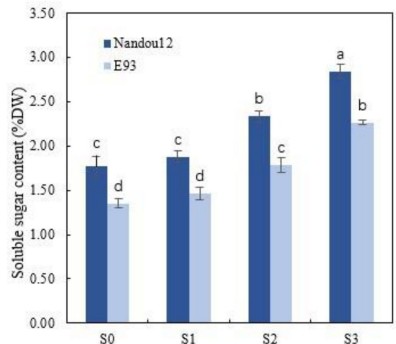

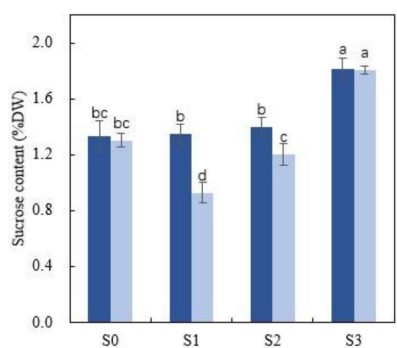

**Figure 4.** *Cont.*

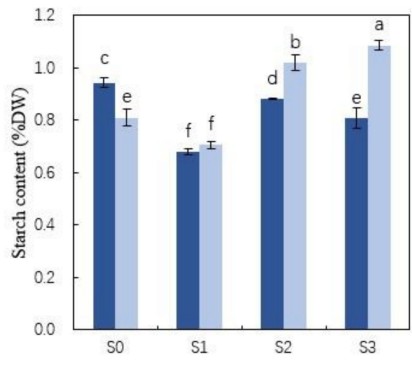

c.

**Figure 4.** Effects of different degrees of shade on the content soluble sugar (**a**), sucrose (**b**) and starch (**c**) of two cultivars of soybean. Different letters indicate statistically significant differences between treatments at $p < 0.05$ by Duncan's multiple range test. Values are mean ± SD ($n = 3$).

### 3.5. Effects of Shading Intensity on Structural Carbohydrates in Stem

The lignin content of the two soybean cultivars was highest under treatment S1, increased by 18.76% and 7.99%, respectively, compared with the control S0; however, lignin content under S2 was lower than that under S0 (S1 > S0 > S2 > S3; Figure 5a). Moreover, lignin content of 'Nandou12' was lower than that of 'E93' under S0 and S1 but higher than that of 'E93' under S2 and S3. Changes in cellulose content displayed a different trend. Cellulose contents under treatments S1, S2 and S3 were significantly lower than those in the control group S0 (Figure 5b). The cellulose content under S1 was higher than those under S2 and S3 (Figure 5b); however, the difference was not significant in 'Nandou12'. In addition, the cellulose content of 'Nandou12' was higher than that of 'E93' under all treatments.

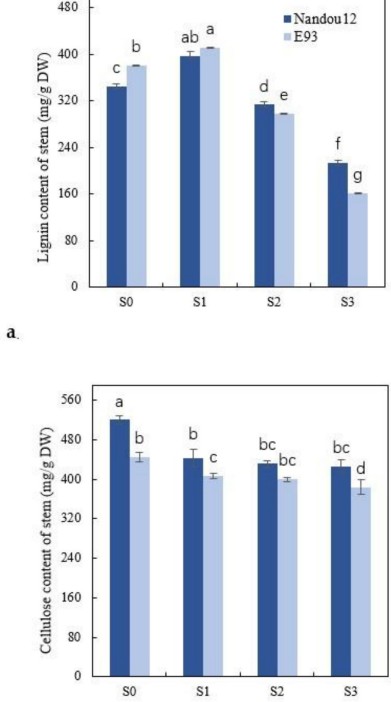

a.

b.

**Figure 5.** Effects of different degrees of shade on the content of lignin (**a**) and cellulose (**b**) of two cultivars of soybean. Different letters indicate statistically significant differences between treatments at $p < 0.05$ by Duncan's multiple range test. Values are mean ± SD ($n = 3$).

### 3.6. Effects of Shading Intensity on Enzymes Related to Lignin Biosynthesis

There was no significant difference in PAL activity between the two cultivars under S1 treatment or in the control group S0; however, with rising shade level, PAL activity decreased considerably (S1 > S2 > S3; Figure 6a). PAL activity of 'Nandou12' in each group was observably higher than that of 'E93'. The 4CL activity of the two cultivars under S1 treatment was considerably higher than that under S0 (by 6.89% and 14.12%, respectively), but the activity under S2 and S3 was significantly lower than that of the control group (Figure 6b). Furthermore, the 4CL activity of 'E93' under each shade treatment was higher than that of 'Nandou12', while the two cultivars showed a similar trend. By contrast, there was a difference in POD activity between the two cultivars (Figure 6c). With increasing shade, the POD activity of 'Nandou1' decreased significantly; the activity of this enzyme under the three shade treatments was lower than that under S0, while the difference between S1 and S2 was not significant. However, the POD activity of 'E93' was highest under the slight shade treatment S1, being 6.68% higher than that under S0. The POD activity of 'E93' under all three shade treatments was greater than that of 'Nandou12'. Except for the severe shade treatment S3, there was no remarkable difference in the CAD activity of 'Nandou12' under the different treatments, while the CAD activity of 'E93' under S1 and S2 was noticeably higher than that under S0 (Figure 6d). In addition, the activity of CAD in 'Nandou12' under S0 was greater than that of 'E93'; there was no significant difference between CAD activities of the two cultivars in the other three groups.

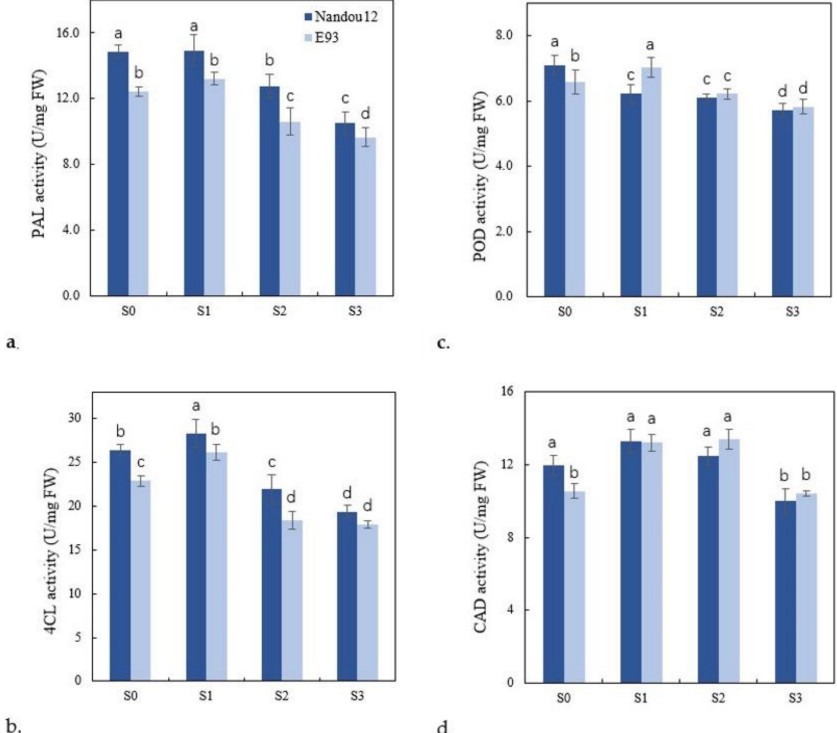

**Figure 6.** Effects of different shades on key enzymes of lignin synthesis in two soybean cultivars. PAL: phenylalanine ammonia-lyase (**a**); 4CL: 4-coumaric acid ligase (**b**); POD: peroxidase (**c**); CAD: cinnamyl alcohol dehydrogenase (**d**). Vertical bars represent ± standard deviation (*n* = 3). S0: no shade, S1: slight, or 43% reduction in light intensity, S2: moderate, or 58% reduction, S3: heavy, or 73% reduction.

### 3.7. Effects of Shading Intensity on Key Lignin Biosynthetic Genes in Soybean Stems

*LAC*, *C3H* and *4CL* gene expression levels in 'Nandou12' increased and then decreased with increasing shade intensity (Figure 7b, Table 2). Compared with S0, *LAC*, *C3H* and *4CL* expression levels of 'Nandou12' under S1 increased by 23.53%, 31% and 2.8%, respectively. Furthermore, *LAC* and *C3H* expression levels of 'Nandou12' under S2 also increased, but by less than that under treatment S1.

The *C3H* gene of 'E93' was significantly upregulated under S1 compared with S0, while *LAC* gene expression was not remarkably changed. When shading intensity was further increased (S2 and S3), *LAC*, *C3H* and *4CL* expression levels in 'E93' were all downregulated compared with those under S0.

Expression levels of *PAL* and *C4H* genes in both 'Nandou12' and 'E93' were not different under S1 or S0. When the shading intensity was low, expression of these two genes in the two cultivars remained stable, and there was no significant difference between them. However, *PAL* and *C4H* expression in 'Nandou12' was gradually downregulated as shading intensity increased (S2 and S3). The expression of these two genes in 'E93' declined sharply with increasing shade, suggesting that 'E93' is more sensitive to shade. Moreover, gene expression levels in 'Nandou12' were higher than those in 'E93'. Variation in *PAL* gene expression was supported by the PAL enzyme data.

*CAD* and *COMT* gene expression in the two cultivars ware similar under different shade treatments. Increasing shading level caused a decrease in *CAD* and *COMT* expression genes, but the expression level of *COMT* was higher in 'Nandou12' than that in 'E93' under each treatment. It is worth noting that expression levels of these two genes under S0 were not notably down-regulated.

Expression of the *CCoAOMT* gene in both cultivars was downregulated under shade treatment; the expression in 'Nandou12' declined gradually with increasing shade level, while that of 'E93' decreased, then recovered and then decreased again. The expression of *CCoAOMT* in 'E93' was significantly lower than that in 'Nandou12' under treatment S1.

Expression of the *CCR* gene in the two cultivars also showed a similar trend, being down-regulated with increasing shade intensity. However, by contrast with the other genes, *CCR* expression in 'E93' under all treatments was higher than that in 'Nandou12'.

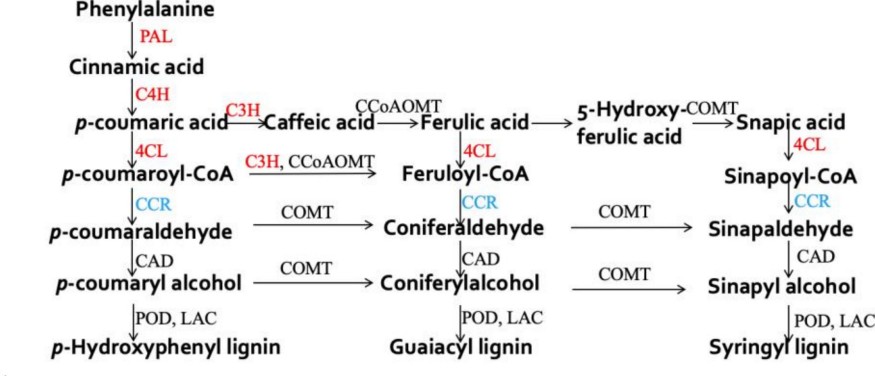

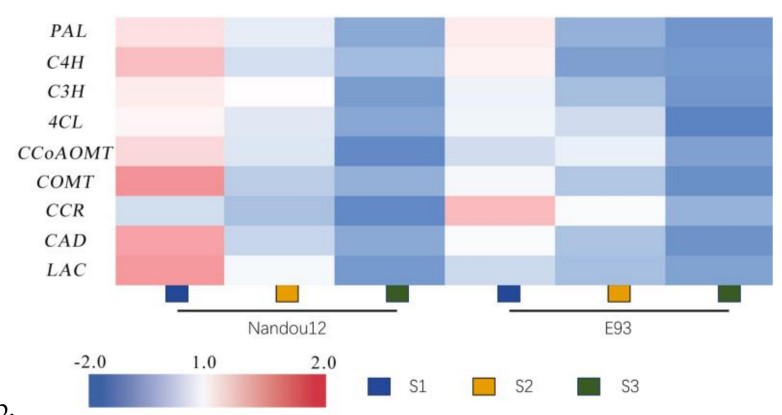

**Figure 7.** (**a**) The pathway of Lignin biosynthesis of soybean, red: up regulated under S1, blue: down-regulated under S1, black: no significant change under S1. (**b**) The heat map analysis of the expression z-scores of several lignin synthetic key genes under different shading levels. The heat map was created by the Adobe Illustrator. The expression level from low (L) to high (H) indicates the minimum and maximum in the entire database.

**Table 2.** Z-scores of lignin synthesis related gene expression in S1, S2 and S3 of two soybean cultivars.

| | Nnadou12 | | | E93 | | |
|---|---|---|---|---|---|---|
| **Gene** | **S1** | **S2** | **S3** | **S1** | **S2** | **S3** |
| *PAL* | 1.16 | 0.63 | −0.86 | 1.11 | −0.69 | −1.34 |
| *C4H* | 1.35 | 0.35 | −0.46 | 1.07 | −1.10 | −1.22 |
| *C3H* | 1.10 | 1.01 | −1.16 | 0.75 | −0.40 | −1.30 |
| *4CL* | 1.05 | 0.55 | −0.92 | 0.79 | 0.27 | −1.75 |
| *CCoAOMT* | 1.21 | 0.49 | −1.60 | 0.29 | 0.69 | −1.08 |
| *COMT* | 1.59 | −0.06 | −0.71 | 0.86 | −0.20 | −1.47 |
| *CCR* | 0.32 | −0.34 | −1.61 | 1.36 | 0.94 | −0.68 |
| *CAD* | 1.50 | 0.15 | −0.88 | 0.93 | −0.29 | −1.41 |
| *LAC* | 1.55 | 0.88 | −1.24 | 0.22 | −0.37 | −1.04 |

Z-scores were performed using Excel software.

## *3.8. Correlation Analysis*

As can be seen from Table 3, lodging resistance index and breaking strength were remarkably correlated with lignin content in both soybean cultivars. Although the correlation between xylem proportion and the two indexes of lodging resistance was not significant, it was relatively high, exceeding 0.5. However, there were some differences between the two cultivars. Lignin biosynthesis-related enzymes of 'Nandou12' were more strongly correlated with lodging resistance index and breaking strength than those of 'E93'. Three of the key enzymes (PAL, 4CL and POD) were significantly correlated with breaking strength, while two (PAL and POD) were considerably correlated with lodging resistance index; we observed no obvious correlations in 'E93'.

**Table 3.** The correlation between lodging resistance and lignin synthesis of two soybean cultivars under different shade degrees.

| | | LRI | BS | LC | XP | PAL | 4CL | POD | CAD |
|---|---|---|---|---|---|---|---|---|---|
| | LRI | 1 | | | | | | | |
| | BS | 0.982 ** | 1 | | | | | | |
| | LC | 0.748 * | 0.822 * | 1 | | | | | |
| Nandou12 | XP | 0.69 | 0.778 * | 0.54 | 1 | | | | |
| | PAL | 0.803 * | 0.898 * | 0.846 * | 0.904 * | 1 | | | |
| | 4CL | 0.726 | 0.827 * | 0.686 | 0.980 ** | 0.967 * | 1 | | |
| | POD | 0.988 ** | 0.971 * | 0.814 * | 0.609 | 0.786 * | 0.673 | 1 | |
| | CAD | 0.366 | 0.531 | 0.763 * | 0.687 | 0.832 * | 0.794 | 0.385 | 1 |
| | LRI | 1 | | | | | | | |
| | BS | 0.999 ** | 1 | | | | | | |
| | LC | 0.824 * | 0.820 * | 1 | | | | | |
| E93 | XP | 0.692 | 0.681 | 0.973 * | 1 | | | | |
| | PAL | 0.686 | 0.698 | 0.932 * | 0.899 * | 1 | | | |
| | 4CL | 0.546 | 0.568 | 0.823 * | 0.788 * | 0.972 * | 1 | | |
| | POD | 0.563 | 0.572 | 0.904 * | 0.912 * | 0.982 * | 0.964 * | 1 | |
| | CAD | −0.185 | −0.212 | 0.346 | 0.55 | 0.304 | 0.257 | 0.462 | 1 |

Pearson's correlation and significance analysis were performed using SPSS software. LRI: lodging resistance index; BS: breaking strength; LC: lignin content; XP: xylem proportion. * Significant at 5% level ($p = 0.05$); ** significant at 1% level ($p = 0.01$), $n = 4$.

## 4. Discussion

High-intensity shading can cause lodging of soybean, with plant height and stem diameter representing important morphological indexes [25]. The taller the plant and the thinner the stem, the more prone it is to lodging [26]. Several studies have shown that shade can cause stem elongation and decrease in stem diameter, stem-breaking strength and lodging resistance index; moreover, these changes are related to cultivar and shading intensity [27]. Our experimental results showed that

plant height of the two soybean cultivars increased significantly under all shade levels, consistent with previous conclusions. However, it is worth mentioning that the stem diameter under S1 did not decrease considerably, but remained consistent with that under S0. For this reason, the stem breaking strength of two cultivars did not decrease significantly under slight shade, and that of 'E93' increased slightly compared with that under S0. Lodging resistance index is an important indicator of crop lodging resistance, which is composed of plant main stem length, stem diameter, stem-breaking strength, and aboveground biomass [28]. This is calculated as (stem-breaking strength × stem diameter)/(aboveground dry weight × main stem length) [29]. Although stem diameter was not reduced under treatment S1, increasing plant height caused lodging resistance index to decrease. However, the two cultivars showed different degrees of decline in lodging resistance index; the reduction in 'Nandou12' compared with S0 was less than that in 'E93' and was not significant, possibly because 'Nandou12' is a shade-tolerant cultivar that performs better than 'Nandou12' under the same shade conditions.

Xylem is a transport tissue and also the support structure of plants. Studies have shown that xylem area and xylem proportion are remarkably positively correlated with lodging resistance [30], while pith area and pith proportion are negatively correlated with lodging resistance [31]. Therefore, we measured these indicators. The results showed that slight shade significantly increased the xylem proportion of the two soybean cultivars, while pith ratio showed no difference. This means that under a condition of invariable stem diameter, a greater proportion of the stem of a 'Nandou12' plant is xylem under light shade compared to S0 (control), which provides the plant with greater support and prevents lodging. 'E93' showed no corresponding differences owing to genotypic and shade-tolerance differences between the two cultivars.

Non-structural carbohydrates in plants mainly include soluble sugar, sucrose and starch, among which soluble sugar and sucrose are the main utilization and transportation forms [32] while starch is the main storage form of photosynthetic products [33]. According to our data, S1 treatment observably reduced the starch content of soybean stems, but the soluble sugar and sucrose remained at the same level as in the control group. Under S2 and S3 shade treatments, the content of these unstructured carbohydrates was significantly higher than that in the control group. This may be because plant photosynthates are insufficient due to the heavy shade, and most of the photosynthates are involved in energy supply; plants under slight shade showed little difference from the control group, indicating that growth was not affected. The same results were obtained in wheat. Research of Xu et al. showed that slight shade did not significantly affect photosynthetic product accumulation in wheat [34]. A study on sweet potato showed that slight shade has little effect on plant growth and photosynthesis, and even has some positive effects [35].

Structural carbohydrates mainly include lignin, which is the main component of cell walls, and cellulose, which is abundant in the xylem of plants [36]. Structural carbohydrate is closely related to stem strength. Although cellulose content was also significantly correlated with lodging resistance, there was no significant trend in the change of cellulose content under different shade intensities. Therefore, we mainly focused on the effect of lignin and its biosynthesis on the mechanical strength of stems. The results of correlation analysis showed that lodging resistance index was significantly positively correlated with breaking strength, which was confirmed by calculating lodging resistance index. Previous studies showed that severe shading can affect some enzymes related to lignin biosynthesis, such as 4CL, CAD and PAL, as well as the expression level of their biosynthetic genes, so as to reduce the biosynthesis of lignin and affect lodging resistance [5,37]. Correlation analysis also showed that the key enzymes of lignin biosynthesis (PAL, POD, 4CL) were significantly correlated with lignin content, xylem proportion and lodging resistance. In this experiment, the lignin content of the two soybean cultivars under S2 and S3 treatments decreased considerably compared to that under S0, with S3 < S2, consistent with the results of previous studies; meanwhile the lignin content of the two cultivars under S1 increased dramatically compared to that under S0. The enzymes and

genes involved in lignin biosynthesis also suggested that slight shade had little effect on the lignin biosynthesis pathway.

Under S1, the activity of some key enzymes in lignin biosynthesis, such as PAL, 4CL and POD, was not remarkably different from that under S0. POD is the last enzyme in lignin biosynthesis, which catalyzes dehydrogenation polymerization of various woody alcohol monomers [38]. CAD activity was improved by participating in and regulating lignin polymerization in the cell wall. Some studies have shown that CAD activity is positively correlated with lignin content [17,39]. Moreover, the expression levels of key genes related to lignin biosynthesis showed only minimal differences under S1 treatment compared with those under S0. These may have been caused by the slight shading stress at the seedling stage. Specifically, there were some differences between the two cultivars: 'Nandou12' generally performed better than 'E93' and maintained maximum intracellular physiological stability under slight shade, which was related to its cultivar characteristics. Although lignin-related enzymes and genes were strongly inhibited under heavy shade, the degree of inhibition was nearly the same for the two cultivars, suggesting that despite differences in shade tolerance between the two cultivars, these differences narrowed when shade intensity reached a certain level.

From our study, finally, we suggest a schematic diagram that slight and heavy shade had different effects on lodging resistance of soybean. Slight shade had no or insignificant effect on the key enzymes and related genes of lignin synthesis, therefore, there was no remarkable change in lignin content and xylem proportion in soybean stems. For this reason, the lodging resistance of soybean was not considerably reduced. In contrast, under severe shade, the strong inhibition of lignin synthesis eventually caused a prominent decrease in lodging resistance of soybean stem. (Figure 8).

Some studies showed that abiotic stress at the seedling stage can improve the tolerance of plants to corresponding adversity at a later stage [40,41]. After the removal of the shading net, the shade between plants gradually intensified due to soybean growth. Shading in the early stage improved adaptability to shading of treatment S1, and plants thus showed higher stem lignin content than those under S0 without shading. In the maize-soybean intercropping system, the symbiotic period of the two crops is short, and the maize is harvested when the soybean is in the seedling stage. Shading during the intercropping period, therefore, plays a role in training the soybean seedlings, so the quality of intercropped soybean is better [42]. However, there was no noteworthy difference in cellulose content between the two cultivars under different shade intensities.

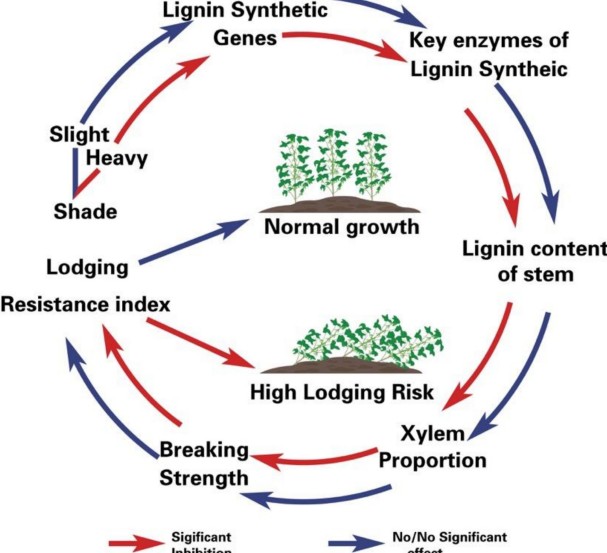

**Figure 8.** Schematic diagram of the effect of shade degree on lodging resistance and lignin synthesis of the stem in soybean.

## 5. Conclusions

Shade significantly affected stem strength and lodging resistance, which were related to soybean genotype and shade level. The effect of shade intensity on the content of cellulose in stems was not prominent, while the lignin content was different under the four shade treatments. Slight shading of seedlings increased the level of lignin in the stem, although this was dependent on cultivar. Lignin level was positively correlated with lodging resistance in shaded soybeans. In addition, slight shade had little effect on the activity of the rate-limiting enzymes PAL, 4CL, POD and CAD in the lignin biosynthesis pathway of the two soybean cultivars, and the activity of 4CL even increased. The expression levels of most of the key genes in the lignin biosynthesis pathway were not considerably down-regulated by slight shade, while *C3H*, *4CL* and *LAC* were upregulated compared with the natural light treatment. Specifically, there were some differences between the two cultivars; 'Nandou12' generally performed better and maintained maximum intracellular physiological stability under slight shade, which was related to its cultivar characteristics, compared to E93, Nando12 is a shade-tolerant cultivar. In other words, the difference in lignin synthesis between the two soybean cultivars may also be one of the reasons for their different shade tolerance.

**Supplementary Materials:** The following are available online at http://www.mdpi.com/2073-4395/10/4/544/s1, Table S1: Genes and primers used for quantitative reverse transcription (RT)-PCR analysis.

**Author Contributions:** Conceptualization, B.W. and W.L.; methodology, Y.Z.; software, X.Z.; validation, B.W., S.W. and J.Y.; formal analysis, M.X., S.Q.; resources, W.L.; data curation, B.W.; writing—original draft preparation, B.W. and S.H.; writing—review and editing, B.W. and W.L.; supervision, W.Y.; project administration, W.L.; funding acquisition, W.L. All authors have read and agreed to the published version of the manuscript.

**Funding:** This work was funded by the National Natural Science Foundation of China (31671626 and 31201170).

**Acknowledgments:** We thank International Science Editing (http://www.internationalscienceediting.com) for editing this manuscript.

**Conflicts of Interest:** The authors declare no conflict of interest. The funders had no role in the design of the study; in the collection, analyses, or interpretation of data; in the writing of the manuscript, or in the decision to publish the results.

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
