# Peer review of "Slight Shading Stress at Seedling Stage Does not Reduce Lignin Biosynthesis or Affect Lodging Resistance of Soybean Stems"

_agronomy, doi:10.3390/agronomy10040544_

Round 1
Reviewer 1 Report
This article gives important and interesting information in crop production, however, huge revision, including Discussion and Conclusion, is needed. The revision points are as follows;
- At least, a couple of year needed at this kinds of open field pot experiment. Otherwise, it is needed that present of regular environmental conditions, like daily weather and temperature, and regular watering (irrigation), although 3 replications and comparative data analyses were executed.
- Checking some of sentences is needed.
*Lines 47~49: However, like some other non-shade plants, the growth and development of 47 soybean in the shade environment can be affected to some extent, the most significant manifestation 48 is prone to lodging.
-However, Like some other non-shade plants, even though? the growth and development of soybean in the shade environment can be affected to some extent, the most significant manifestation is prone to lodging.
*Lines 289~294: Are these sentences consist of just one sentence?
Lodging resistance index is the important indicators of crop lodging resistance, which composed of the plant main stem length, stem diameter, stem breaking strength and the aboveground biomass[22], the formula for [(stem breaking strength×stem diameter)/(aboveground dry weight×main stem length)][23], although the stem diameter not reduce under S1, but due to the rising of plant height makes the lodging resistance index decreased, however, two cultivars has different degree of decline, the reduction of Nandou12 is less than that of E93, and compared with S0 did not reach significant level, this may be due to the Nandou12 is a shade resistance cultivar of reasons, so it performs better under the same shade condition.
- Abstract: Some sentences are not logical.
*lines 18~20: The activity of lignin synthesis rate-limiting enzymes PAL, POD and CAD, which 18 were considerably related to these two lodging resistance indexes, was not significantly decreased 19 by slight shade, moreover, the activity of 4CL was also increased.
-The activity of lignin synthesis rate-limiting enzymes PAL, POD and CAD, which 18 were considerably related to these two lodging resistance indexes, was not significantly decreased by slight shade, while moreover, the activity of 4CL was also increased.
*lines 24~29: It is very difficult to accept the suggestion that ‘high yield and quality can be achieved by setting the applicable shade level’ at this experiment.
-lines 24 to 29: These physiological and molecular changes resulted in differences in the stem strength performance of soybean under different shade intensities, there were also some specific physique among soybean cultivars.
This means that high yield and quality can be achieved by setting the applicable shade level, in addition, exploiting the lodging resistance potential of existing soybean cultivars is also an effective and efficient way to address the yield reduction caused by lodging in intercropped soybeans.
- Materials and methods
-Description of genes and primers used in R-T PCR (Table S1) is needed in Materials and methods (2.3.6 Gene expression).
- Results
*Table 2 and its explanation must moved to ‘Results’ part.
*Supplement of ‘Stem anatomical structure’ photo plate: Representative photo plates to fit Table 1 are needed.
- Discussion and Conclusion;
-Description at ‘Discussion’ and ‘Conclusion’ parts is not fit with the ‘Discussion and Conclusion.’
*Discussion:
1) ‘Some studies~(lines 345~346; 348~350)’ needed plural references.
2) Figure 7 : Move to ‘Discussion’ part and correct description of Fig. 7 is needed.
3) Supplement of ‘slight shading effect’ in other crops must be needed.
*Conclusion:
-Most of the description in ‘Conclusion’ part have to move to ‘Discussion’ part, and reconstruct ‘Conclusion’ part.
- Details on pages1to 9 are given by attached file.

Author Response
Response to Reviewer 1 Comments
Dear Reviewer,
Thank you for your letter and the reviewer’s comments concerning our manuscript. These comments are valuable and helpful for revising and improving our paper. We have studied the comments carefully and made corrections, which we hope will meet with your approval. The main corrections to the manuscript and our responses to the reviewers’ comments are given below.
Point 1:
At least, a couple of year needed at this kinds of open field pot experiment. Otherwise, it is needed that present of regular environmental conditions, like daily weather and temperature, and regular watering (irrigation), although 3 replications and comparative data analyses were executed.
Response 1:
Thanks for your advice, because the environment and soil conditions were well controlled, only one year's data was used, while we have added data about daily temperature and humidity in the methods section. (Figure 1)
Point 2:
Checking some of sentences is needed.
*Lines 47~49: However, like some other non-shade plants, the growth and development of 47 soybean in the shade environment can be affected to some extent, the most significant manifestation 48 is prone to lodging.
-However, Like some other non-shade plants, even though? the growth and development of soybean in the shade environment can be affected to some extent, the most significant manifestation is prone to lodging.
Response2:
Thanks for your advice, we have modified this sentence in the manuscript. (L44-46)
Point 3:
*Lines 289~294: Are these sentences consist of just one sentence?
Lodging resistance index is the important indicators of crop lodging resistance, which composed of the plant main stem length, stem diameter, stem breaking strength and the aboveground biomass[22], the formula for [(stem breaking strength×stem diameter)/(aboveground dry weight×main stem length)][23], although the stem diameter not reduce under S1, but due to the rising of plant height makes the lodging resistance index decreased, however, two cultivars has different degree of decline, the reduction of Nandou12 is less than that of E93, and compared with S0 did not reach significant level, this may be due to the Nandou12 is a shade resistance cultivar of reasons, so it performs better under the same shade condition.
Response 3:
Thanks for your helpful suggestion, we have broken this long sentence into a few sentences. (L322-330)
Point 4:
Abstract: Some sentences are not logical.
*lines 18~20: The activity of lignin synthesis rate-limiting enzymes PAL, POD and CAD, which 18 were considerably related to these two lodging resistance indexes, was not significantly decreased 19 by slight shade, moreover, the activity of 4CL was also increased.
-The activity of lignin synthesis rate-limiting enzymes PAL, POD and CAD, which 18 were considerably related to these two lodging resistance indexes, was not significantly decreased by slight shade, while moreover, the activity of 4CL was also increased.
Response 4:
Thank you for your advice. We have modified it according to your suggestion. (L16-20)
Point 5:
*lines 24~29: It is very difficult to accept the suggestion that ‘high yield and quality can be achieved by setting the applicable shade level’ at this experiment.
-lines 24 to 29: These physiological and molecular changes resulted in differences in the stem strength performance of soybean under different shade intensities, there were also some specific physique among soybean cultivars.
This means that high yield and quality can be achieved by setting the applicable shade level, in addition, exploiting the lodging resistance potential of existing soybean cultivars is also an effective and efficient way to address the yield reduction caused by lodging in intercropped soybeans.
Response 5:
Thank you for your advice. We removed the inappropriate description and changed it to “These physiological and molecular changes suggested that higher stem mechanical strength and lodging resistance can be achieved by employing applicable shade levels.” (L25-27)
Point 6:
Materials and methods
-Description of genes and primers used in R-T PCR (Table S1) is needed in Materials and methods (2.3.6 Gene expression).
Response 6:
Thanks for your attention and suggestion. We have added a description of gene names and primers in the methods section.(L141-144)
Point 7:
Results
*Table 2 and its explanation must moved to ‘Results’ part.
Response 7:
We have moved the table to the results section and further refined the description of the table. (L296-304, Table 3)
Point 8:
*Supplement of ‘Stem anatomical structure’ photo plate: Representative photo plates to fit Table 1 are needed.
Response 8:
Thanks for your suggestion. We have inserted pictures of stem anatomical structure into the manuscript. (Figure 3a-j)
Point 9:
Discussion and Conclusion;
-Description at ‘Discussion’ and ‘Conclusion’ parts is not fit with the ‘Discussion and Conclusion.’
*Discussion:
1) ‘Some studies~(lines 345~346; 348~350)’ needed plural references.
Response 9:
Thanks for your suggestion. We have added more references in this article.
Point 10:
2) Figure 7: Move to ‘Discussion’ part and correct description of Fig. 7 is needed.
Response 10:
Thank you for your attention. Actually, figure 7 (Figure 8) was inserted incorrectly, we moved it to “Discussion” in the manuscript. (Figure 8)
Point 11:
3) Supplement of ‘slight shading effect’ in other crops must be needed.
Response11:
Thanks for your suggestion. We included in the discussion the effects of slight shade on winter wheat and sweet potatoes. (L348-351)
Point 12:
*Conclusion:
-Most of the description in ‘Conclusion’ part have to move to ‘Discussion’ part, and reconstruct ‘Conclusion’ part.
Response 12:
Thanks for your advice. We have moved the discussion statement to the "Discussion" section and reconstructed the "Conclusion" section. (L376-382)
Reviewer 2 Report
My response to the manuscript is generally positive.
It is a satisfactory set of experiments to investigate shading in soybean. There are several english edits throughout the manuscript that need to be addressed.
For example, line# 75 "We hope to conduct a series of experiment...".
Also the authors should take more care to using their on writing. For example there are some text in the introduction line#57 that has some similarity from the text of Goujon et al 2003.
Author Response
Response to Reviewer 2 Comments
Dear Reviewer,
Thank you for your letter and the comments concerning our manuscript. These comments are valuable and helpful for revising and improving our paper. We have studied the comments carefully and made corrections, which we hope will meet with your approval. The main corrections to the manuscript and our responses to the reviewers’ comments are given below.
Point 1:
It is a satisfactory set of experiments to investigate shading in soybean. There are several english edits throughout the manuscript that need to be addressed.
For example, line# 75 "We hope to conduct a series of experiment...".
Response 1:
Thanks for your advice. We have sent this article to a professional English editing company for editing.
[International Science Editing ( http://www.internationalscienceediting.com )]
Point 2:
Also the authors should take more care to using their on writing. For example there are some text in the introduction line#57 that has some similarity from the text of Goujon et al 2003.
Response 2:
Thanks for your attention and suggestion. We have revised the inappropriate sentences in the article and marked them in red. We hope to meet your requirements.
Reviewer 3 Report
This is a relatively straightforward analysis of the effect of shading on the growth of soybean plants. I found the data to be sound, though not that exciting. The main challenge is that the grammar is not at publication quality. Many of the concepts are hard to interpret and evaluate because of the wording of the sentences. I think this manuscript could be prepared for publication in this or another journal with significant editing of the manuscript focused on clarity and readability.
Author Response
Response to Reviewer 3 Comments
Dear Reviewer,
Thanks for your letter and the comments concerning our manuscript. These comments are valuable and helpful for revising and improving our paper. We have studied the comments carefully and made corrections, which we hope will meet with your approval. The main corrections to the manuscript and our responses to the reviewers’ comments are given below.
Point 1:
This is a relatively straightforward analysis of the effect of shading on the growth of soybean plants. I found the data to be sound, though not that exciting. The main challenge is that the grammar is not at publication quality. Many of the concepts are hard to interpret and evaluate because of the wording of the sentences. I think this manuscript could be prepared for publication in this or another journal with significant editing of the manuscript focused on clarity and readability.
Response 1:
Thanks for your advice.
We have sent this article to a professional English editing company for editing.[International Science Editing ( http://www.internationalscienceediting.com )]
We have revised the inappropriate sentences and incorrect grammar in the article and marked them in red. We hope to meet your requirements.
Reviewer 4 Report
This study investigates the physiological, transcription and enzymatic activity, and correlation of traits associated with shade tolerance in two soybean cultivars. The detailed measurements taken and analyses performed were impressive. I commend this team for taking the extensive time and effort to perform these experiments in the crop of interest. It is particularly relevant that the studies utilized cultivars which are important for the southwest region of China. However, to fully integrate these findings into the scientific cannon of soybean shade tolerance requires additional work as the presentation of the findings is lacking. This is mostly due to a lack of comparison and integration of previous shade studies in soybean. I believe the overall findings of this study will eventually prove informative and interesting.
The manuscript needs to be read by a native English speaker. Throughout the manuscript extra words and phrases muddle the scientific findings and significance.
The first time all abbreviations are used (like PAL, CAD, 4CL, and POD), they should be preceded by the full name with the abbreviation then provided within parentheses. Though this paper focuses on lignin, only abbreviations are provided for important enzymes and genes discussed. Without the full name of these, readers are unable to put this information in context.
In the abstract, the authors have designated four shade treatments, but then characterize heavy shading by inhibition of specific genes and enzymes. This is using your results to justify your setup. I don’t think that’s what the researchers did, I think they’re trying to explain their results in the context of the shade, but this needs to be re-written to better reflect that.
Ln 26: The phrase ‘there were also some specific physique among soybean cultivars does not make sense.
Ln 42: Shade not only affects grain yield and quality’ – this hasn’t been brought up in the manuscript yet.
Ln 52: ‘After Cellulose’ – you need to address how cellulose is affected by shading before you can use this phrase. If cellulose is affected by shading, please discuss or present logic on why this discussion is not warranted.
Ln 60: you can’t use enzymatic abbreviations like ‘PAL, CAD, 4CL, and POD’ without first presenting the full name of the enzymes in the MS.
Ln 64: if the rice genotypes were shade resistant then shouldn’t the enzymatic activities be higher, not limiting, given the statements in lines 62-64?
Ln 86: How many pots per treatment?
Ln 94: every soybean plant is ‘planted’ until mature should be ‘maintained until mature’.
Figures 1 and 2 should be combined into a single figure.
Table 2: Given the relationships between the area and proportions, especially for pith, I think presenting the data as a bar graph would be more impactful.
Figure 6: If all these enzyme are in the same pathway, the pathway should be diagramed and the expression levels of genes within the pathway depicted as colors. Further, since the expression of all genes are normalized against S0 and with actin, then an actual scale should be provided. Alternatively, the expression levels could be transformed into Z-scores. Either of those options would be acceptable, the current figure is unacceptable with just L vs H.
Figure 7 can be eliminated.
I’m concerned by how few references are cited within the manuscript. Lodging and shade tolerance are both well studied traits in soybean. Without incorporating how these findings fit with previous studies, it’s unclear what the novel findings (if any) of this study are.
Table 2: I like the correlations, but it is unclear what data was used to generate them. Are these correlations for a single timepoint or all the data for both genotypes? Should at least be separated by genotype – which would still allow all data to be presented in a single table.
Ln 350: ‘After the removal of shading net’… when was the shade net removed? I understood the plants were maintained under shade conditions to maturity (line 94). If shade was only a temporary then this needs to be better explained in the materials and methods, especially since the analyses were performed on mature plants. If the netting was removed then these findings could be even more agriculturally relevant and exciting since stress treatments early in the plant life have profound effects on mature plants.
Ln 366 is not a complete sentence. Also, you shouldn’t introduce syrene acrylic acid in the conclusion. If it’s important, introduce and discuss it earlier so you can focus on its importance in the conclusion.
Single plants per pot were grown to maturity. It’s surprising that no analysis is presented on the effect of shading on either yield or seed composition, the two most important factors in soybean production. Especially since line 42 directly references the impact of shade on grain quality. If this data was collected, please include it.
Author Response
Response to Reviewer 4 Comments
Dear Reviewer,
Thank you for your letter the comments concerning our manuscript. These comments are valuable and helpful for revising and improving our paper. We have studied the comments carefully and made corrections, which we hope will meet with your approval. The main corrections to the manuscript and our responses to the reviewers’ comments are given below.
Point 1:
The manuscript needs to be read by a native English speaker. Throughout the manuscript extra words and phrases muddle the scientific findings and significance.
Response 1:
Thanks for your advice. We have sent this article to a professional English editing company for editing.
[International Science Editing ( http://www.internationalscienceediting.com )]
Point 2:
The first time all abbreviations are used (like PAL, CAD, 4CL, and POD), they should be preceded by the full name with the abbreviation then provided within parentheses. Though this paper focuses on lignin, only abbreviations are provided for important enzymes and genes discussed. Without the full name of these, readers are unable to put this information in context. Ln 60: you can’t use enzymatic abbreviations like ‘PAL, CAD, 4CL, and POD’ without first presenting the full name of the enzymes in the MS.
Response 2:
Thank you for your suggestion. We added the full names of all abbreviations that first appeared (L17-18, L19-20, L141-144).
Point 3:
In the abstract, the authors have designated four shade treatments, but then characterize heavy shading by inhibition of specific genes and enzymes. This is using your results to justify your setup. I don’t think that’s what the researchers did, I think they’re trying to explain their results in the context of the shade, but this needs to be re-written to better reflect that.
Response 3:
Thanks for your suggestion, we have rewritten this part of abstract with more objective statements. (L22-25)
Point 4:
Ln 26: The phrase ‘there were also some specific physique among soybean cultivars does not make sense.
Response 4:
Thank you very much for your suggestion. We have deleted this sentence from the manuscript. (L29)
Point 5:
Ln 42: Shade not only affects grain yield and quality’ – this hasn’t been brought up in the manuscript yet.
Response 5:
Thanks for your attention and suggestion. We have deleted this sentence from the manuscript. (L42)
Point 6:
Ln 52: ‘After Cellulose’ – you need to address how cellulose is affected by shading before you can use this phrase. If cellulose is affected by shading, please discuss or present logic on why this discussion is not warranted.
Response 6:
Thanks for your advice. Since cellulose is not the focus of this article and the experiment is not carried out around cellulose, so we rewrote this paragraph. (L48)
Point 7:
Ln 64: if the rice genotypes were shade resistant then shouldn’t the enzymatic activities be higher, not limiting, given the statements in lines 62-64?
Response 7:
Thank you very much for pointing this out, we have modified this sentence to be more accurate. (L66-68)
Point 8:
Ln 86: How many pots per treatment?
Response 8:
Thank you very much for your suggestion, it is very important. Each treatment has 30 pots, which we supplemented in the manuscript. (L95-96)
Point 9:
Ln 94: every soybean plant is ‘planted’ until mature should be ‘maintained until mature’.
Response 9:
Thank you very much for providing more accurate wording, which we have modified in the article. (L98)
Point 10:
Figures 1 and 2 should be combined into a single figure.
Response 10:
We have combined the two figures as you suggested and modified the description of it in the article. (Figure 2)
Point 11:
Table 2: Given the relationships between the area and proportions, especially for pith, I think presenting the data as a bar graph would be more impactful.
Response 11:
Thanks for your attention, it is really a useful suggestion, we have redone the table and added the bar graphs of pith and xylem proportion. (Figure 3a, b)
Point 12:
Figure 6: If all these enzymes are in the same pathway, the pathway should be diagramed and the expression levels of genes within the pathway depicted as colors. Further, since the expression of all genes are normalized against S0 and with actin, then an actual scale should be provided. Alternatively, the expression levels could be transformed into Z-scores. Either of those options would be acceptable, the current figure is unacceptable with just L vs H.
Response 12:
Thank you very much for such useful advice. We have added z-scores of gene expression levels and diagram of lignin synthesis pathways that contain all the genes in the results. (Figure 7a, Table 2)
Point 13:
Figure 7 can be eliminated.
Response 13:
Actually, figure 7 (Figure 8) was inserted incorrectly, it should be placed after discussion or as a figure abstract, depending on the editor's choice. We moved the insertion position of it in the manuscript. (Figure 8)
Point 14:
I’m concerned by how few references are cited within the manuscript. Lodging and shade tolerance are both well studied traits in soybean. Without incorporating how these findings fit with previous studies, it’s unclear what the novel findings (if any) of this study are.
Response 14:
Thanks for your Suggestions, we have inserted more references in the discussion section to better support our conclusion.
Point 15:
Table 2: I like the correlations, but it is unclear what data was used to generate them. Are these correlations for a single timepoint or all the data for both genotypes? Should at least be separated by genotype – which would still allow all data to be presented in a single table.
Response 15:
Thanks for your advice. We have redone the table of correlation analysis according to two different soybean cultivars, and we have also added some discussion about genotype differences in the article. (Table 3)
Point 16:
Ln 350: ‘After the removal of shading net’… when was the shade net removed? I understood the plants were maintained under shade conditions to maturity (line 94). If shade was only a temporary then this needs to be better explained in the materials and methods, especially since the analyses were performed on mature plants. If the netting was removed then these findings could be even more agriculturally relevant and exciting since stress treatments early in the plant life have profound effects on mature plants.
Response 16:
Yes, we remove the shading net before soybean flowering (R1). We also described our point of view in the discussion section. We believe that whether it is harvesting intercropping maize first or removing the shading net before soybean reproductive growth, soybean can get some exercise in the early stage. We have described the removal time of shading net more clearly in the method section. (L99)
Point 17:
Ln 366 is not a complete sentence. Also, you shouldn’t introduce syrene acrylic acid in the conclusion. If it’s important, introduce and discuss it earlier so you can focus on its importance in the conclusion.
Response 17:
Thanks for your suggestion, we have moved this part of description to the introduction. (L58-63)
Point 18:
Single plants per pot were grown to maturity. It’s surprising that no analysis is presented on the effect of shading on either yield or seed composition, the two most important factors in soybean production. Especially since line 42 directly references the impact of shade on grain quality. If this data was collected, please include it.
Response 18:
Thanks for your advice and attention. This paper only discusses the relationship between soybean lodging and light environment. Yield and quality are not the focus of this paper. For this reason, there's no data of yield or seed composition.
Round 2
Reviewer 1 Report
Comments by reviewer were mostly revised except explanation of Figure 8. Other fine revisions were checked on the attached file.

Author Response
Response to Reviewer 1 Comments
Dear Reviewer,
Thank you for your letter and the more detailed suggestions concerning our manuscript. These Suggestions are very important to raise the level of this article. According to these advices, we have made modifications and marked them in the manuscript, which we hope will meet with your approval. Our responses to the comments are given below.
Point 1:
Explanation for Figure 8.
Response 1:
Thanks for your advice. We have rewritten the explanation of figure 8 as your suggestion, we hope it will meet with your approval. (L399-405)
Point 2:
Latin names, formatting issues, etc.
Response 2:
Thanks for your attention, we have modified all the issues according to your attachment file. (please see the track changes in the article)
Reviewer 3 Report
The revised manuscript is much better. Thanks for making the grammar corrections.
I'm good with everything except a few claims in the Abstract and conclusions that need to be corrected.
The Abstract states that:
"These physiological and molecular changes suggested that higher stem mechanical strength and lodging resistance can be achieved by employing applicable shade levels."
This comes across as suggesting that application of shade would result in higher stem mechanical strength and lodging resistance. Your paper indicates exactly the opposite. Please revise or remove.
I disagree with the statement in the conclusion:
"The expression levels of most of the key genes in the lignin biosynthesis pathway were not considerably down-regulated by shade, while C3H, 4CL and LAC were upregulated compared with the natural light treatment."
The data shows that all of them are downregulated under heavy shade. Please remove or correct.
Author Response
Response to Reviewer 3 Comments
Dear Reviewer,
Thank you for your letter and the more detailed suggestions concerning our manuscript. These Suggestions are very important to raise the level of this article. According to these advices, we have made modifications and marked them in the manuscript, which we hope will meet with your approval. Our responses to the comments are given below.
Point 1:
The Abstract states that:
"These physiological and molecular changes suggested that higher stem mechanical strength and lodging resistance can be achieved by employing applicable shade levels."
This comes across as suggesting that application of shade would result in higher stem mechanical strength and lodging resistance. Your paper indicates exactly the opposite. Please revise or remove.
Response 1:
Thanks for your suggestion, we have revised it more accurately in the article. (L25-27)
Point 2:
I disagree with the statement in the conclusion:
"The expression levels of most of the key genes in the lignin biosynthesis pathway were not considerably down-regulated by shade, while C3H, 4CL and LAC were upregulated compared with the natural light treatment."
The data shows that all of them are downregulated under heavy shade. Please remove or correct.
Response 2:
Thank you for your attention and useful suggestion, we have corrected it in the “Conclusion”. (L427-429)
Reviewer 4 Report
I greatly appreciate the efforts the authors undertook to improve the english language of the manuscript. The paper was much easier to read and I was able to spend more time thinking about the science. I found the changes the authors made greatly improved the manuscript. However, there are still a few changes that must be incorporated prior to publication.
1. Both cultivars are shade tolerant? In lines 86-88, the cultivars are described as Nandou12...strong shade tolerance and E93..a shade tolerant cultivar. A better explanation of why two shade tolerant cultivars were chosen for this study is warranted.
2. I really like the inclusion of Figure 3. However, it's difficult to compare the images as they are all in different orientations.
3. Figure 7 and Table 2 present the same information. I strongly believe a heat map best conveys the information being presented, however, as I indicated in my previous report, the current heat map is unacceptable. The expression scale can not be H and L. Since the expression of all genes are normalized against S0 and with actin, then an actual scale should be provided. Alternatively, if the authors want to present a heat map of the z-scores (data presented in table 2), that would also be acceptable. Additionally, the heat map should be organized in the same order as the metabolic pathway. The current order doesn't reflect any biological relevance or information.
4. Figure 7a, the lignin biosynthesis pathway. Thank you for including this. Please clarify the color coding of the enzymes. The colors should reflect the up or down regulation of the enzymes in the two genotypes.
5. Table 2. One of the cultivars is labeled Nannong-996, I believe you mean E93.
6. The conclusion presents a nice summary of the findings. However, it would have more impact if this could be extended to the biology underlying shade tolerance for the two shade-tolerant cultivars.
Author Response
Response to Reviewer 4 Comments
Dear Reviewer,
Thank you for your letter and the more detailed suggestions concerning our manuscript. These Suggestions are very important to raise the level of this article. According to these advices, we have made modifications and marked them in the manuscript, which we hope will meet with your approval. Our responses to the comments are given below.
Point 1:
Both cultivars are shade tolerant? In lines 86-88, the cultivars are described as Nandou12...strong shade tolerance and E93..a shade tolerant cultivar. A better explanation of why two shade tolerant cultivars were chosen for this study is warranted.
Response 1:
Thanks for your attention, our description of E93 may not accurate. E93 is a control cultivar selected in our previous study, and it is a shade intolerance cultivar, for this reason, we choose E93 as the control cultivar. We have modified it in the article. (L88)
Point 2:
I really like the inclusion of Figure 3. However, it's difficult to compare the images as they are all in different orientations.
Response 2:
Thank you for your affirmation and suggestion, we have adjusted these pictures to the same orientation as far as possible. We hope to meet your requirements. (Figure 3c-j)
Point 3:
Figure 7 and Table 2 present the same information. I strongly believe a heat map best conveys the information being presented, however, as I indicated in my previous report, the current heat map is unacceptable. The expression scale can not be H and L. Since the expression of all genes are normalized against S0 and with actin, then an actual scale should be provided. Alternatively, if the authors want to present a heat map of the z-scores (data presented in table 2), that would also be acceptable. Additionally, the heat map should be organized in the same order as the metabolic pathway. The current order doesn't reflect any biological relevance or information.
Response 3:
Thanks very much for your suggestions, we have changed the Figure 7b to the heat map of z-scores, in addition, we have reordered the gene names (both in Fig.7b and Table 2) in the order of the lignin synthesis pathway.
Point 4:
Figure 7a, the lignin biosynthesis pathway. Thank you for including this. Please clarify the color coding of the enzymes. The colors should reflect the up or down regulation of the enzymes in the two genotypes.
Response 4:
Thanks for your advice, we have modified the Figure 7a. Because the regulation of the lignin biosynthesis genes of two soybean cultivars under S0-S3 is different, and this paper is mainly focus on the slight shade, so we have labeled the colors according to the regulation under S1. (red: up regulated under S1, blue: down-regulated under S1, black: no significant change under S1)
Point 5:
Table 2. One of the cultivars is labeled Nannong-996, I believe you mean E93.
Response 5:
Thanks for your attention. Yes, it means E93, we have modified it in Table 2.
Point 6:
The conclusion presents a nice summary of the findings. However, it would have more impact if this could be extended to the biology underlying shade tolerance for the two shade-tolerant cultivars.
Response 6:
Thanks for your useful advice. We have added a description of the relationship between the biology underlying shade tolerance of two genotypes and the results of lignin synthesis. (L431-433)